# Antimicrobial Resistance Pattern of *Escherichia coli* Isolates from Small Scale Dairy Cattle in Dar es Salaam, Tanzania

**DOI:** 10.3390/ani12141853

**Published:** 2022-07-21

**Authors:** Rogers R. Azabo, Stephen E. Mshana, Mecky I. Matee, Sharadhuli I. Kimera

**Affiliations:** 1Department of Veterinary Microbiology, Parasitology and Biotechnology, College of Veterinary Medicine and Biomedical Sciences, Sokoine University of Agriculture, P.O. Box 3019, Morogoro 67125, Tanzania; 2National Livestock Resources Research Institute, Nakyesasa, Kampala P.O. Box 5704, Uganda; 3SACIDS Africa Centre of Excellence for Infectious Diseases, Sokoine University of Agriculture, P.O. Box 3297, Morogoro 67125, Tanzania; mateemecky@gmail.com (M.I.M.); sikimera@gmail.com (S.I.K.); 4Department of Microbiology and Immunology, Weill Bugando School of Medicine, Catholic University of Health and Allied Sciences, P.O. Box 1464, Mwanza 33109, Tanzania; stephen72mshana@gmail.com; 5Department of Microbiology and Immunology, School of Medicine, Muhimbili University of Health and Allied Sciences, P.O. Box 65001, Dar es Salaam 11103, Tanzania; 6Department of Veterinary Medicine and Public Health, College of Veterinary Medicine and Biomedical Sciences, Sokoine University of Agriculture, P.O. Box 3021, Morogoro 67125, Tanzania

**Keywords:** antimicrobial resistance, cattle, *Escherichia coli*, Dar es Salaam, Tanzania

## Abstract

**Simple Summary:**

Dearth of information on antimicrobial resistance (AMR) in small-scale dairy cattle in Dar es Salaam, the commercial city of Tanzania, prompted us to conduct this study. The objective was to determine the different levels of resistance phenotypical patterns among *Escherichia coli* (*E. coli*) isolates from rectal swabs of apparently healthy cattle. Antimicrobial resistance occurs when microorganisms develop the ability to tolerate antimicrobial concentrations to which they were initially susceptible. It is a phenomenon of global concern, which is on the rise due to antimicrobial use in food-producing animals. In dairy farms, cattle carry high levels of AMR *Escherichia coli* (*E. coli*), and may act as a potential reservoir. The study revealed that resistance to ampicillin, cefotaxime, tetracycline and trimethoprim/sulfamethoxazole was the most frequent. Resistance to nalidixic acid, ciprofloxacin, chloramphenicol, and gentamycin was also observed among the *E. coli* isolates, but with lower percentages. *E. coli* resistant to third generation cephalosporins was also detected. The results of the current study give an insight into the status of antimicrobial resistance and multidrug resistance in small-scale dairy cattle in Dar es Salaam, Tanzania. The findings call for further research, prudent antimicrobial use, and surveillance initiatives.

**Abstract:**

In Tanzania, information on antimicrobial resistance in small-scale dairy cattle is scarce. This cross-sectional study was conducted to determine the different levels and pattern of antimicrobial resistance (AMR), in 121 *Escherichia coli* isolated from rectal swab of 201 apparently healthy small-scale dairy cattle in Dar es Salaam, Tanzania. Isolation and identification of *E. coli* were carried out using enrichment media, selective media, and biochemical tests. Antimicrobial susceptibility testing was carried out using the Kirby–Bauer disk diffusion method on Mueller-Hinton agar (Merck), according to the recommendations of Clinical and Laboratory Standards Institute (CLSI). Resistance was tested against ampicillin, gentamicin, chloramphenicol, trimethoprim-sulfamethoxazole, tetracycline, nalidixic acid, ciprofloxacin and cefotaxime. Resistance to almost all antimicrobial agents was observed. The agents to which resistance was demonstrated most frequently were ampicillin (96.7%), cefotaxime (95.0%), tetracycline (50.4%), trimethoprim-sulfamethoxazole (42.1%) and nalidixic acid (33.1%). In this case, 20 extended-spectrum beta-lactamases (ESBLs) producing *E. coli* were identified. 74.4% (90/121) of the isolates were Multidrug resistant (MDR), ranging from a combination of three to 8 different classes. The most frequently observed phenotypes were AMP-SXT-CTX with a prevalence of 12.4%, followed by the combination AMP-CTX with 10.7% and TE-AMP-CTX and NA + TE + AMP + CTX with 8.3% each. The high prevalence and wide range of AMR calls for prudent antimicrobial use.

## 1. Introduction

Tanzania is currently ranked as Africa’s second largest source of livestock [1]. Livestock subsector is an important and integral part of Tanzania’s agriculture and a key source of national food security [2]. Cattle are one of the most important livestock species in terms of yield and capital value [2]. Based on the 2016/2017 Livestock Sector Analysis (LSA) report [2], the national herd was 28.4 million. However, according to the latest census of agriculture (livestock) in 2019/2020, the cattle population is 33.9 million [1], an increase of 19.4%. The increasing need for dietary protein for animal-derived foods has led to intensive livestock production in which the use of antimicrobials is unavoidable [3]. Intensive animal husbandry production is widely adopted by both small and large producers as it is considered a more viable way to ensure global food security [4]. In Tanzania, antimicrobials are widely used in the livestock industry for therapeutic and prophylactic purposes as well as growth promotion [5]. Previous studies have confirmed the improper use of antibiotics in cattle in both intensive and large-scale production by herders and nomads [6,7,8]. Indiscriminate use may be due to the availability of antibiotics without a veterinary prescription and the widespread trade of veterinary medicines [7]. The most commonly used antibiotics in livestock are tetracyclines, penicillins, aminoglycosides, macrolides, and sulphonamides [5,8]. 

Antibiotics have significantly reduced infection-related mortality, however high-dose and repetitive use in animal production is the most important in the selection, emergence and dissemination of antimicrobial resistance (AMR). There is evidence that it is an important factor [9,10]. It also led to the development of bacterial multidrug resistance (MDR) to these drugs [11]. Some of the practices associated with antibiotic abuse are incorrect treatment plans, incorrect diagnoses, incorrect medication, or non-compliance with the required treatment time [12]. AMR has emerged as one of the major threats to human and animal health [13,14], and animal production has been identified as one of the hotspots associated with the emergence and spread of antimicrobial resistance [15,16]. This justifies the need for continuous monitoring of antimicrobial-resistant strains in food producing animals, in order to understand and mitigate AMR. AMR poses a threat to the health and existing drug stockpiles for treating infectious diseases in humans and animals [17,18]. High levels of antimicrobial resistance have been reported in food animals in African countries, with resistance to penicillin and tetracycline being the most frequently observed [19,20].

Antibiotics are commonly used to target pathogenic organisms. However, they also exert selective pressure on commensal gut bacteria, thereby promoting the development and maintenance of antimicrobial resistance in these bacteria [21]. Both antibacterial resistant pathogens and commensal organisms can reach humans through direct contact with animals or animal products [22] or through consumption of animal products [23,24]. Studies have shown that the selection of resistant bacteria can take place at sub-minimum inhibitory concentrations, facilitating the occurrence of antimicrobial resistance [25,26]. Despite the abundant resistance phenotypes observed in the bacteria, their mechanism of action is a specific gene on any mobile genetic elements such as plasmids, transposons, and integrons, which are either transmitted vertically or horizontally [27]. Recently, Tanzanian authorities established a comprehensive national action plan on antibiotic resistance, year 2017 to 2022, to tackle resistance nationwide including animal husbandry among their targets.

*Escherichia coli* (*E. coli*), though a commensal bacterium of the intestinal flora of humans and animals, they are commonly used as indicator organisms for antimicrobial resistance [28]. This is because they are extensively distributed in the gut and easily acquire those genes that encode antimicrobial resistance due to their genomic plasticity [29]. Therefore, it serves as a reservoir for lifelong antimicrobial resistance genes and exerts pressure on the intestinal flora of the organism exposed to the pressures applied on the gut flora of the organism [30]. 

In Tanzania, several studies have been conducted on poultry and pig production [31,32,33,34,35,36,37]. However, there is dearth of information on antimicrobial resistance profiles of isolates from healthy small-scale dairy cattle to antibiotics commonly used in humans and livestock. The recent sharp rise in antibiotic usage in livestock from 176,932 kgs in 2010 to 419,957 kgs in 2017 in Tanzania [38] and Dar es Salaam region from 55.3 tonnes in 2016 to 60.4 tonnes in 2018 [39] calls the urgent need for monitoring of antimicrobial use in small scale cattle production. Such information is required by policymakers to guide the prudent use of antimicrobials in animal production.

## 2. Materials and Methods

### 2.1. Ethical Considerations

The Medical Research Coordinating Committee of the National Institute for Medical Research (NIMR) of Tanzania gave ethical approvals for (Reference No. NIMR/HQ/R.8a/Vol. IX /3233) and Sokoine University of Agriculture (Permit No. DPRTC/186/3). The permission was sought from the relevant authorities which are the municipal directors of the two districts. Then verbal consent from the participating farmers.

### 2.2. Study Design, Area and Cattle Farm Selection

This was a cross-sectional study conducted from September to October 2021 in Kinondoni and Ubungo districts, which are part of the city of Dar es Salaam in eastern, Tanzania. It included five wards: Kijitonyama, Kunduchi and Wazo in Kinondoni and Mbezi and Saranga in Ubungo (Figure 1). The selection of cattle farms included in this study was randomized based on a list of cattle herders provided by the local livestock officers in the study area. However, the selection was not totally random, as verbal consent from the herd owners was needed. 54 farms with a total herd of 201 participated in the study. As part of the city, most small-scale dairy farms had up to four heads of cattle as stipulated by the city bylaws, but some farms had more than 10 heads of cattle. Of the 54 small-scale dairy cattle farms, 39 farms each had four heads of cattle, all of which were sampled. The rest of the farm had more than ten heads of cattle. At each of the latter farms, 1 to 3 three heads of cattle were randomly selected and sampled according to the milk yield. In almost all of these 15 farms, cattle were high producers, so only low-yield cattle were allowed to be sampled.

### 2.3. Sample Collection

Briefly, cattle were restrained in a pen. Using gloved hands, the tail was raised using one hand and the other for swabbing. Approximately one gram of cattle faecal material was aseptically swabbed from the rectum using a sterile cotton swab (Himedia, Mumbai, India). The swab was then placed in sterile tubes filled with 3 mL Stuart’s transport medium, cap tightened, and shaken well. All samples were transported in a clean cool box with ice packs at a temperature of 2 to 8 °C and processed within 48 h of collection at the Microbiology Teaching Laboratory at the College of Veterinary Medicine and Biomedical Sciences, Sokoine University of Agriculture (SUA), Morogoro, Tanzania.

### 2.4. Bacterial Isolation and E. coli Identification

Looped rectal swab specimens from Stuart transport medium, were streaked directly onto the MacConkey agar (Oxoid, Basingstoke, UK) without antibiotics and aerobically incubated at 37 °C for 24 h. On incubation, for the case of mixed growth, a single colony from almost morphologically similar colonies (deep pinkish, round, mid-sized, and flat) was selected for the purity-plate, from each plain MacConkey agar plate and aerobically sub-cultured on nutrient agar medium (Hi Media, Mumbai, India) at 37 °C for 18–24 h, as previously reported [37]. Pure colonies were gram stained and subjected to biochemical tests to identify *E. coli* according to the manufacturer’s instructions using indole methyl red, Voges–Proskauer, and citrate utilization test (IMViC; Sigma-Aldrich Co., Zurich, Switzerland). Isolates that indicated indole positive, methyl-red positive, Voges–Proskauer negative, and citrate negative were identified as *E. coli* [40]. The identified *E. coli* isolates were subjected to an antibiotic susceptibility test (AST) and phenotypic confirmation of ESBL production.

### 2.5. Antimicrobial Susceptibility Testing

Antimicrobial susceptibility test was performed using the Kirby–Bauer disk diffusion method on Mueller-Hinton agar (Merck), according to the recommendations of the Clinical and Laboratory Standards Institute (CLSI) [41]. The *E. coli* isolates were tested using the following antibiotics shown in Table 1. 

These antibiotics have been selected because are considered to be clinical and useful in animal production by the World Health Organization (WHO) [42]. Single well-isolated colonies from the identified pure lactose fermenter cultures were emulsified into 5 mL of sterile saline and adjusted to that of standard 0.5 McFarland (Oxoid Ltd., Basingstoke, UK). A sterile cotton swab was dipped in a standardized bacterial (*E. coli*) suspension and evenly streaked or spread over the entire surface of Mueller Hinton agar medium. Impregnated paper discs with constant concentration of antibiotics were placed on the surface of the agar and incubated at 37 °C for 24 h. The zone of inhibition interpretation was performed according to the CLSI 31st Edition guidelines [41]. All the *E. coli* isolates that showed intermediate susceptibility to the antibiotics tested were not regarded as resistant to such particular antibiotics. A double-disk synergistic test of cefotaxime and cefotaxime-clavulanate was used to identify the extended-spectrum ß-lactamases (ESBL) phenotype [41]. *E. coli* ATCC 25922 was used as a control strain. Percentages of isolates resistant to any number of antibiotics were reported. If the isolate was resistant to three or more drugs from different classes of antimicrobials, it was considered MDR [43].

### 2.6. Statistical Analysis

Statistical analysis was performed using IBM SPSS Statistics for windows version 20.0. The graphical representation was performed using a program (Microsoft Office Excel, 2019). The Kruskal–Wallis test was used with a significance value of 0.05 to compare the co-resistance of different isolates. This is a non-parametric statistical test that evaluates differences between three or more independently sampled groups on a single, non-normally distributed continuous variable [44].

## 3. Results

This study evaluated the antimicrobial resistance of *E. coli* isolates from healthy cattle rectal swabs. A total of 201 rectal swab samples were examined and 121 *E. coli* isolates were isolated. This corresponds to an isolation rate of 60.2%.

### 3.1. Antimicrobial Susceptibiliy

The prevalence of antimicrobial resistance in 121 *E. coli* isolates at the isolate level *is* presented in Table 2. 

The majority of isolates (Table 2 and Figure 2) have shown resistance to ampicillin (96.7%), followed by cefotaxime (95.0%), tetracycline (50.4%), trimethoprim/sulfamethoxazole (42.1%) and nalidixic acid (33.1%). Moderate resistance rates were observed for ciprofloxacin (20.7). However, low resistance was observed with chloramphenicol (19.0%) and gentamycin (10.7%). In this case, 20 extended-spectrum beta-lactamases (ESBLs) that produce *E. coli* were detected (Table 3). 

#### 3.1.1. Coresistances and AMR Profile

Analysis of coresistance for the *E. coli* isolates is shown in Table 4. 

A Kruskal-Wallis H test showed that there was no statistically significant difference in isolates across the categories of resistance, x^2^ = 3.049, *p* = 0.384, with a mean rank isolate of 19.50 for category of resistance to One antimicrobial agent, 15.80 for category of resistance to Two antimicrobial agents, 26.43 for category of resistance to Three antimicrobial agents and 19.88 for category of resistance to More than three agents.

On the other hand, 90 isolates (74.4%) were MDR, which means resistant to three or more families of antibiotics, as we considered beta-lactams and cephalosporins as two different families. 

#### 3.1.2. MDR Patterns and Antimicrobial Resistance Phenotype

Depending on their antibiotic resistance phenotypes, the 121 isolates of *E. coli* belong to 41 different phenotypes, showing a large variety of resistances, ranging from one antimicrobial to a combination of 8 (Table 5).

The most frequently observed phenotypes were AMP-SXT-CTX with a prevalence of 12.4%, followed by AMP-CTX (10.7%), TE-AMP-CTX and NA + TE + AMP + CTX with a prevalence of 8.3% each. In this case, 15 different phenotypes that produce extended-spectrum ß-lactamases (ESBLs) were detected and had a prevalence of 42.9%. They were all MDR ranging from three antimicrobials to a combination of 8. ESBL phenotypes were resistant to ampicillin and cefotaxime.

## 4. Discussion

In this study, 60.2% of the isolates from 201 samples of rectal swabs from small scale dairy cattle were recovered. Overall, 74.4%, which is close to three quarters of all *E. coli* isolates from the small-scale dairy cattle, exhibited multidrug resistance against three to eight classes of antimicrobial, with the most resistant pattern being AMP-SXT-CTX. The 121 *E. coli* isolates belonged to 41 phenotypes ranging from one antimicrobial agent to a combination of 8. 20 *E. coli* isolates were ESBLs producers with a prevalence rate of 16.5%. There were 15 different phenotypes detected with a prevalence of 42.9% producing extended-spectrum ß-lactamases and all of them were MDR ranging from three antimicrobials to a combination of 8. The occurrence of a high-level antibiotic resistance obtained in *E. coli* isolates from the healthy dairy cattle studied gives an indication of the possible roles these bacteria play in the transfer of the observed resistance among pathogenic *E. coli* from livestock in Tanzania. The resistance levels identified in this study are comparable with the findings reported by other studies in *E. coli* isolates from healthy cattle elsewhere in Africa. For instance, Ogunleye et al. [45] reported 96.7% (232/240) resistance for ampicillin among *E. coli* isolated from rectal swab from apparently healthy cattle from a major cattle market in Ibadan, Oyo State, South West Nigeria, 59.1% (117/198) from rectal swab from healthy cattle in Eastern Algeria, Barour et al. [46], 54.8% (23/42) from apparently healthy cattle in greater part of Kumasi, Ghana, Rita et al. [47], while in the current study, 96.7% (117/121) resistance was obtained for ampicillin. However, the resistance percentage reported in this study was higher than the 21.4% (29/135) in Madoshi et al. [48] in Tanzania.

It was expected from the start of the study for a possibility of at least observing some level of resistance to cephalosporins. A study by Katakweba et al. [49] reported some level of resistance by cefotaxime (57.1% (40/70), a 3rd generation cephalosporin, in Enterobacteriaceae isolated from dairy cattle in Tanzania. The percentage resistance in the current study for cefotaxime were also comparable to the findings of Olowe et al. [50] in Nigeria. Cefotaxime had 95.0% (115/121) resistance in the current study and 92.1% (105/114) as reported by Olowe et al. [50]. However, it was in contrast with the findings of Barour et al. [46] who reported a low resistance rate of 4.5% (9/198). Cephalosporins are not readily available for veterinary use and are rarely used on small scale in farms in Tanzania [48]. Surprisingly, the current study showed widespread occurrence of resistance to cephalosporins in *E. coli* from farm animals. This could be explained by cross transmission of resistant strains between cattle and humans where there is widespread use of cephalosporin Although in this study, we did not address antibiotic use in humans, the exchange of resistance genes between humans and animals is likely to occur during the rainy season when cattle drinks contaminated water or pastures. Therefore, the 95.0% resistance to cefotaxime could imply a potentially serious threat to public health. Remarkable resistance in the current study was also observed against tetracycline and trimethoprim/sulfamethoxazole which is comparable to the findings of Abbassi et al. [51] in Tunisia but lower than the findings of Gupta et al. [52] in Bangladesh. For tetracycline, there was 50.4% (61/121) in the current study; 33.3% (20/60) for Abbassi et al. [51] and 83.3% (70/84) for Gupta et al. [52]. For trimethoprim/sulfamethoxazole, there was 42.1% (51/121) resistance in the current study, 30% (18/60) in Abbasi et al. [51] and 83.3% (70/84) from the work of Gupta et al. [52]. Resistance to tetracycline and trimethoprim/sulfamethoxazole might have relation with the wide use of these antibiotics in livestock production over along period on farms in Tanzania. In the current study, a high resistance level of cefotaxime was observed as compared to that of tetracycline and trimethoprim/sulfamethoxazole. The contrast in cefotaxime resistance is difficult to explain, as the antibiotic was not commonly used on animals. Further studies are probably required to address this observation. However, this antibiotic could possibly serve as an indicator for monitoring drug resistance in animals and humans.

In this study, 16.5% (20/121) of the *E. coli* isolates were found to be ESBLs producers, which is comparable to other studies in Tanzania and elsewhere. For instance, Seni et al. [53], reported 10.8% (14/130) of the *E. coli* isolates from cattle as ESBL producers in Tanzania; Chishimba et al. [54] reported 20.1% (77/384) of *E. coli* isolates in market-ready broiler chicken in Zambia; Reuben et al. [55] reported 10% (20/200) of *E. coli* isolates from cattle in Tanzania; Okpara et al. [56], reported 10.6% (53/457) of *E. coli* isolates from animals and humans in households in Nigeria. However, the prevalence reported in the current study is lower than the levels reported by Montso et al. [57], of 66.3% (69/104) in *E. coli* isolates from cattle on farms and raw beef in South Africa and Kimera et al. [37] of 65.3% (301/461) in broiler chicken in Tanzania. Almost all the ESBL producers were resistant to most of the tested antimicrobials. This is probably indicative of the selection pressure exerted by excessive use of beta-lactams and cephalosporins in animal production, and presence of various resistance mechanisms from inappropriate use of veterinary drugs [58].

The MDR levels in the current study are comparable with the findings reported by other studies in *E. coli* isolates from clinically health cattle elsewhere. For instance, Ajayi et al. [59], reported 81.0% (851/1051) MDR level from *E. coli* isolates from clinically health cattle in Nigeria, 61.1% (99/162) in Nariman et al. [60] from rectal swabs of healthy cattle from the suburban and farming areas in Tripoli, Libya, 64.3% (27/42) in Rita et al. [47] in Kumasi, Ghana from fresh droppings from clinically health cattle at the abattoir, and 44.2% (77/174) in Abbasi et al. [51], in Tunisia, from clinically healthy cattle while in the current study, 74.4% (90/121) MDR level was obtained. However, it was lower than the findings of Donkor et al. [61], in Ghana of 97.7% (262/268). The possible explanation of MDR level in this study could be attributed to indiscriminate exploitation of antimicrobial agents in small scale dairy cattle thus providing selective pressure favouring the emergence of resistant strains. That is to say, the MDR reported in this study can probably be the result of an independent resistance for each antimicrobial agent or coresistance. MDR bacteria presents an impending therapeutic impasse to human and animal health. Some major factors which may contribute to the increase of bacterial MDR are: transfer of resistance determinants by genetic elements such as plasmids, transposons, and gene cassettes into integrons and changing regulation in mar locus [62]. The high MDR level we recorded in the current study calls for monitoring of MDR *E. coli* strains.

The antimicrobial susceptibility patterns observed in the isolates from the small- scale dairy cattle towards one to a combination of 8 antimicrobials in this study shows that the isolates were diverse in their antimicrobial resistance spectrum. This is comparable to the findings of Barour et al. [46], who reported a large variety of resistances, ranging from one antimicrobial to a combination of 10. MDR patterns in the current study were comparable to some studies in *E. coli* isolates from apparently health cattle elsewhere. For instance, Amosun et al. [63] reported 41 MDR patterns from *E. coli* isolates from apparently health on-farm cattle, in Ibadan, Nigeria, 31 MDR patterns in Barour et al. [46], in Algeria from rectal swab *E. coli* isolates from health cattle and 33- MDR patterns in the current study. However, it was lower than the 71 MDR patterns in Ajayi et al. [59] in Ado-Ekiti, Nigeria from *E. coli* isolated from feces of apparently healthy ready to slaughter cattle. MDR patterns may harbor resistance genes that may be transferred to humans via the food chain. There is need for further studies on identifying the resistance genes and their ability to mobilize. All the MDR phenotypes in the current study were AMP resistant. This was comparable with the findings by Barour et al. [46], who reported that all the MDR phenotypes were AMP resistant. This implies that *E. coli* strains resistant to this antibiotic have an increased ability to be resistant to other antimicrobials. 

In this case, 15 out of the 20 ESBL phenotypes in the current study were MDR. This was comparable with the studies in *E. coli* isolates from clinically health cattle carried out elsewhere. For instance, Olowe et al. [50] in Nigeria, reported 63.2% (72/114) as ESBL phenotypes and MDR, 4.5% (9/198) in Barour et al. [46], in Algeria and in the current study was 16.5% (20/121). ESBL phenotype includes resistance to ampicillin and cefotaxime, which is the cause of many therapeutic failures [62]. This requires the surveillance of strains with this type of phenotype.

The high variability of resistance phenotypes in this study can be explained by the acquisition of resistance to several antibiotics of different classes (coresistance) since the plasmids exchanged usually have several resistance genes such as the coresistance of *E. coli* to cephalosporins, penicillins, chloramphenicol, tetracyclines, and fluoroquinolones Barour et al. [46]. 

Despite the above observations, the current study had some limitations. In the first place, there could be other antibiotics used among the small-scale dairy cattle in Tanzania, which were not tested in this study. Secondly, we did not perform molecular characterization of resistant isolates. The high resistance observed could be explained based on standardized Clinical Laboratory Standard Institute (CLSI) 2021 protocol and guidelines [41] which are internationally recognized, thus depicting the real magnitude and pattern of AMR in the study setting. 

## 5. Conclusions

The study has revealed levels of AMR as well as ESBL producers, of *E. coli* isolates from the rectal swab of apparently healthy animals to commonly used antimicrobials in small-scale dairy cattle production. This renders the antimicrobials inactive to their intended use. The results of the current study call for prudent antimicrobial use in cattle to create a better picture of the situation and advice policymakers on their decisions. 

## Figures and Tables

**Figure 1 animals-12-01853-f001:**
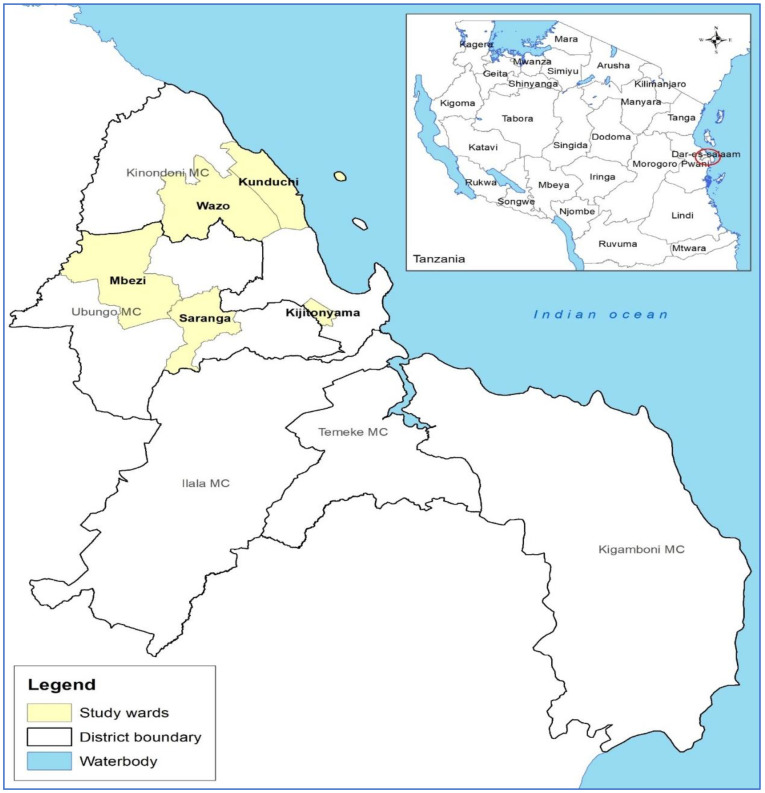
Map of the study districts (wards) in Dar- es- Salaam, Tanzania.

**Figure 2 animals-12-01853-f002:**
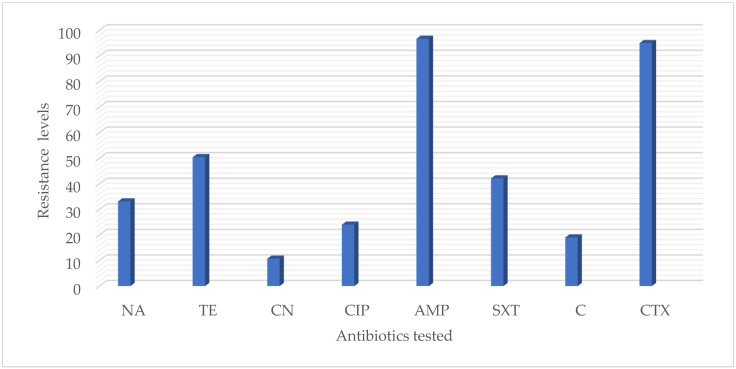
Frequencies of antibiotic resistance in 121 *E. coli* isolates: Nalidixic acid (NA), Tetracycline (TE), Gentamycin (CN), Ciprofloxacin (CIP), Ampicillin (AMP), Trimethoprim/sulfamethoxazole (SXT), Chloramphenicol (C), Cefotaxime (CTX).

**Table 1 animals-12-01853-t001:** Antimicrobial concentrations and interpretation breakpoints of the various antimicrobial agents used in this study to interpret the results (CLSI, 2021).

Antimicrobial Agent (Code)	Disc Drug Concentration (µg)	Breaking Point (mm)		
		Sensitive (S)	Intermediate (I)	Resistant (R)
AMP	10 µg	≥17	14–16	≤13
CTX	30 µg	≥26	23–25	≤22
CN	10 µg	≥15	13–14	≤12
TE	30 µg	≥15	12–14	≤11
NA	30 µg	≥19	14–18	≤13
CIP	5 µg	≥21	16–20	≤15
SXT	1.25/23.75 µg	≥16	11–15	≤10
C	30 µg	≥18	13–17	≤12

AMP: Ampicillin, CTX: Cefotaxime, CN: Gentamycin, TE: Tetracycline, NA: Nalidixic acid, CIP: Ciprofloxacin, SXT: Trimethoprim/sulfamethoxazole, C: Chloramphenicol.

**Table 2 animals-12-01853-t002:** Susceptibility to Antibiotics of *E. coli* Isolates (*n* = 121) from Small-Scale Dairy Cattle in Dar es Salaam, Tanzania.

ATB	R	I	S	Distribution (Number) in Each inhibition zone Diameter (mm)																				
	%	%	%	0	7	8	9	10	11	12	13	14	15	16	17	18	19	20	21	22	23	24	25	>25
NA	33.1	25.6	41.3	33						6	1		3	2	15	11	13	21	1	11	2			2
TE	50.4	7.4	42.1	36	10	9	6			4	3	2	3	8	7	9	7	10		5	1			1
CN	10.7	13.2	76.0	5				4		4	3	13	13	15	17	18	13	14	1	1				
CIP	24.0	26.4	49.6	9	1	1	3	5		2		1	7	5	4	6	7	10	7	12	6	9	10	16
AMP	96.7	1.7	1.7	111	1	1		2			2	1	1		1								1	
SXT	42.1	14.0	43.8	25	3	4	7	12	5	2	5	1	4	6	4	10	10	7		1	5	4	3	3
C	19.0	14.0	66.9	20			1		2		1	4	2	3	7	7	7	20	9	13	7	7	4	7
CTX	95.0	4.1	0.8	25	1		1	2	1	3		3	14	7	6	12	11	20	3	6	2	1	2	1

*E. coli* isolated from plain MacConkey agar. ATB: antibiotic, R: resistance, I: intermediate, S: susceptible, NA: nalidixic acid, TE: tetracycline, CN: gentamycin, CIP: ciprofloxacin, AMP: ampicillin, SXT: trimethoprim/sulfamethoxazole, C: chloramphenicol, CTX: Cefotaxime. Dark grey fields present frequencies of resistant isolates, fields in light grey with borders present frequencies of intermediate isolates, and white fields present frequencies of susceptible isolates.

**Table 3 animals-12-01853-t003:** Phenotypic pattern of 20 ESBL producing *E. coli* isolates from rectal swab of small-scale dairy cattle.

No of Antibiotic Classes	Resistance Pattern	No. of Isolates	Prevalence (%)
2	AMP + CTX	3	15.0
3	AMP + SXT + CTX	2	10.0
	TE + AMP + CTX	3	15.0
	CN + AMP + CTX	1	5.0
4	NA + AMP + SXT + CTX	1	5.0
	NA + TE + AMP + CTX	3	15.0
	TE+ AMP + SXT + CTX	2	10.0
	TE + CIP + AMP + CTX	1	5.0
5	CIP + AMP + SXT + C +CTX	1	5.0
	NA +TE + AMP + SXT + CTX	1	5.0
6	NA + TE + CIP + AMP + C + CTX	2	10

**Table 4 animals-12-01853-t004:** Co-resistance of the *E. coli* isolates from cattle.

	Number (*n*) and Percentages (%) of Isolates Resistant to				Kruskal-Wallis H (*p*-Value)
Categories of resistance to antimicrobial agents	**One**	**Two**	**Three**	**More than three agents**	x^2^ = 3.049 *p* = 0.384 df = 3
	*n* (%)	*n* (%)	*n* (%)	*n* (%)	
	2 (5.0)	5 (12.5)	7 (17.5)	26 (65.0)	
Mean ranks	19.50	15.80	26.43	19.88	

*E. coli*: *Escherichia coli*.

**Table 5 animals-12-01853-t005:** Patterns of antimicrobial resistance phenotypes for *Escherichia coli* strains isolated in the study, with antibiogram pattern codes.

Number of Resistances	Antibiogram Patterns	Codes of Pattern	Number of Isolates
1	AMP	1	3
	CTX	2	1
2	AMP + CTX	3	13
	NA + CTX	4	1
	SXT + CTX	5	1
	TE + CTX	6	1
	CN + AMP	7	1
3	NA + AMP + CTX	8	6
	AMP + SXT + CTX	9	15
	TE + AMP + CTX	10	10
	AMP + C + CTX	11	1
	CN + AMP + CTX	12	3
	CIP + AMP + CTX	13	2
	NA + CIP + AMP	14	1
4	NA + TE + AMP + CTX	15	10
	TE + CIP + AMP + CTX	16	5
	AMP + SXT + C + CTX	17	1
	TE + AMP + SXT + CTX	18	5
	NA + CN + AMP + CTX	19	1
	CN + CIP + AMP + CTX	20	1
	NA + CIP + AMP + CTX	21	2
	TE + CIP + AMP + SXT	22	1
	TE + AMP + C + CTX	23	3
	NA + AMP + SXT + CTX	24	2
	CN + AMP + SXT + CTX	25	1
5	TE + CIP + AMP + SXT + CTX	26	3
	NA + TE+ AMP + SXT+ CTX	27	2
	TE + CIP + AMP + C+CTX	28	1
	TE + CN + AMP + SXT + CTX	29	1
	NA+ CIP + AMP + SXT + CTX	30	3
	TE + AMP + SXT + C + CTX	31	5
	CIP + AMP + SXT + C + CTX	32	1
	NA + TE + CIP + AMP + CTX	33	1
	NA + TE + AMP + C + CTX	34	1
6	TE + CN + AMP + SXT + C + CTX	35	1
	NA + TE + CIP + AMP + C+CTX	37	2
	NA + TE + AMP + SXT + C+CTX	38	3
	NA + TE + CIP + AMP + SXT + CTX	39	2
7	TE + CN + CIP + AMP + SXT + C + CTX	40	1
8	NA + TE + CN + CIP + AMP + SXT + C + CTX	41	3

## Data Availability

The data generated is contained within the article.

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
