# Peer review of "Antimicrobial Resistance Pattern of Escherichia coli Isolates from Small Scale Dairy Cattle in Dar es Salaam, Tanzania"

_animals, 2022, doi:10.3390/ani12141853_

Round 1

Reviewer 1 Report

Line 27-28: "The study revealed that resistance to ampicillin, cefotaxime, tetracycline and trimethoprim /sulfamethoxazole were the most frequent..." add “,” before "and", change "were" to "was".

Line 29: "nalidixic acid, ciprofloxacin, chloramphenicol and gentamycin were also observed among the E. coli" add “,” before "and", change "were" to "was".

Line 46: "identified. 74.4% (90/121) of the isolates were Multidrug resistant (MDR), ranging from combination" change "combination" to "a combination“.

Line 74: remove "the emergence of" to make it simple.

Line 76: add "," before "or"

Line 81-83: "AMR poses a threat to health and existing drug 81 and curb the menace, which poses a threat to the health and existing drug stockpiles for 82 treating infectious diseases in humans and animals". Revise it to make it clear.

Line 101-102: "This is because they are extensively distributed in the gut and easily acquires those  acquires genes that encode antimicrobial resistance due to its their genomic plasticity"

1. There is a sampling bias. Are low yielding cows unhealthy and treated with more antimicrobials?

2. The description of the statistics analysis is too brief, more details need to be provided.

3. From line 345 to 357, the authors associated ESBLs with antibiotic abuse by comparing the results from regions.  To support this conclusion, the authors should provide data on whether there are differences in antibiotic use in these regions.

Author Response

Dear  Reviewer,

Thanks very much for the precious time you put aside to go through manuscript no. 1786305. Whatever was raised was taken in good faith for the improvement of the quality of the manuscript. Attached are the responses to the queries raised.

Regards

Reviewer 2 Report

The manuscript has scientific soundness, showing very important data about the detection of Escherichia coli from dairy cattle. In spite of the study limitation, I believe that authors could work better on the text, showing the relevance of the study. So many manuscripts have been published without any molecular analysis, given that phenotypic analysis guides up to right conclusion about resistance profiles.

Below, some specific comments:

Line 43: from - The agents to which resistance was demonstrated most frequently were ampicillin 43

(96.7%), cefotaxime (95,0%), tetracycline (50.4%) and trimethoprim-sulfamethoxazole (42.1%) and 44

nalidixic acid (33.1%); to - The agents to which resistance was demonstrated most frequently were ampicillin 43

(96.7%), cefotaxime (95,0%), tetracycline (50.4%), trimethoprim-sulfamethoxazole (42.1%) and 44

nalidixic acid (33.1%)

Line 60: The Increasing need to the increasing need...

Line 71: re-write the paragraph, some sentences are redundant and a little confused

Line 156: 37oC for 24 hours 2 to 8 oC .. within 48 h.

Line 209: Antimicrobial susceptabiliy to animicrobial susceptibility

Line 401 - it was not done any molecular analises to support the hypothesis of HGT... So, the sentence should be modified to "could be explained"

Line 406 - you should re-write this paragraph, evidencing the relevance of the data you obtained from your study, "instead of the limitations about the numbers of antibiotic tested and the absence of  samples from small-scale dairy cattle farms where antibiotics are not used since we could not find any due to the widespread use of antimicrobials in animal production in the communities"... 

Tables

I believe that Table 1 is dismissed, since it is find in the CLSI; as well as Table 2, wich can be supressed to only resistance, intermediate susceptibility and susceptibility.

In table 3, instead of One, two, three... agents, it would be more effective in showing these data if the tested antimicrobials and the phenotypic profile for each one of them were included... Thus, it would be possible to visualize the combined resistance profiles.

Table 4. I don't know if the data os statistical analysis needs to be shown in a table; I believe it could be written in the text

Table 5 - excellent

English review is very necessary.

Author Response

(The authors gave the same response as above.)

Reviewer 3 Report

The manuscript's object is to determine the phenotypic resistant profile of E. coli in small-scale cattle farms in Tanzania. The study found resistance to up to eight antimicrobials. The authors conclude a high MDR rate in small cattle farms was alarming and need the policy to mitigate its spread. 

The manuscript is well written and easy to understand. I do not have major comments. Please the following minor edits required in the manuscript.

Line 204. Results. Where is the analysis of the prevalence of antimicrobial resistance of E. coli from different sample locations? Were more isolates collected were resistant? Is there a data difference in AMR prevalence in high yield vs low yield milk cows?

Line 246 -247.  Suggested creating a table or graph to show the phenotypic characteristics of ESBL positive isolates. 

Line 271 - 272. Please provide an additional explanation of why beta-lactam and cephalosporin were separated.

Line 295 - 299. These sentences are confusing regarding the ESBL isolates. Please considered rewriting it to make it easy to understand. 

Line 300 - 302. Would suggest adding the currently available antimicrobials in the local which were readily available and contribute to the existing finding of AMR pattern.

Line 354 - 355. Why the genes encoding to ESBL were not screened? please explain.

Author Response

Dear Reviewer,

Thanks very much for the precious time put aside to go through manuscript no. 1786305. Whatever was raised was taken in good faith for the improvement of the quality of the manuscript. Attached are responses to the queries raised.

Regards

Round 2

Reviewer 1 Report

The authors have responded to the comments.